# LLM Augmented LLMs:
# Expanding Capabilities through Composition

**Rachit Bansal**[1] **Bidisha Samanta**[1] **Siddharth Dalmia**[2] **Nitish Gupta**[1] **Shikhar Vashishth**[1]
**Sriram Ganapathy**[1] **Abhishek Bapna**[1] **Prateek Jain**[1] **Partha Talukdar**[1]
[1]Google Research India    [2]Google DeepMind

## Abstract

Foundational models with billions of parameters which have been trained on large corpora of data have demonstrated non-trivial skills in a variety of domains. However, due to their monolithic structure, it is challenging and expensive to augment them or impart new skills. On the other hand, due to their adaptation abilities, several new instances of these models are being trained towards new domains and tasks. In this work, we study the problem of efficient and practical composition of existing foundation models with more specific models to enable newer capabilities. To this end, we propose CALM—Composition to Augment Language Models—which introduces cross-attention between models to compose their representations and enable new capabilities. Salient features of CALM are: (i) Scales up LLMs on new tasks by 're-using' existing LLMs along with a few additional parameters and data, (ii) Existing model weights are kept intact, and hence preserves existing capabilities, and (iii) Applies to diverse domains and settings. We illustrate that augmenting PaLM2-S with a smaller model trained on low-resource languages results in an absolute improvement of up to 13% on tasks like translation into English and arithmetic reasoning for low-resource languages. Similarly, when PaLM2-S is augmented with a code-specific model, we see a relative improvement of 40% over the base model for code generation and explanation tasks—on-par with fully fine-tuned counterparts.

## 1 Introduction

Large Language Models (LLMs) have shown to encompass a range of foundational capabilities such as commonsense and factual reasoning, world knowledge, and coherent language generation (Bubeck et al., 2023; Google et al., 2023). Leveraging these foundational capabilities, a number of efforts in the community have fine-tuned these models to enable domain-specific capabilities such as code generation, copy editing, and mathematical problem solving (Lewkowycz et al., 2022; Singhal et al., 2023). This has resulted in the development of several specialized large models with domain-specific capabilities. For example, there are models that do well on standard code generation but are not as proficient in general logical reasoning and vice-versa. Presence of such a large number of domain-specific models leads to a natural question: Can we compose an *anchor* model with a domain-specific *augmenting* model to enable new capabilities? For example, can we compose an augmenting model's code understanding capability with an anchor LLM's language generation capability to enable code-to-text generation capability?

The typical approach for this problem is to further pre-train or (efficiently) fine-tune the anchor model on the data that was originally used to train the augmenting model (Hu et al., 2021; Kessler et al., 2022). However, many a times such solutions are not feasible since training large models is computationally expensive, especially since the augmenting model itself may be an LLM trained on a massive corpora. Further, processing data from multiple sources might not be feasible due to privacy concerns and organizational boundaries. Working with multiple distinct models is also desirable since it allows the reuse of existing models with established capabilities, providing better control and avoiding catastrophic forgetting that is prevalent in conventional approaches.

---

Correspondence to Rachit and Bidisha: [brachit, bidishasamanta]@google.com

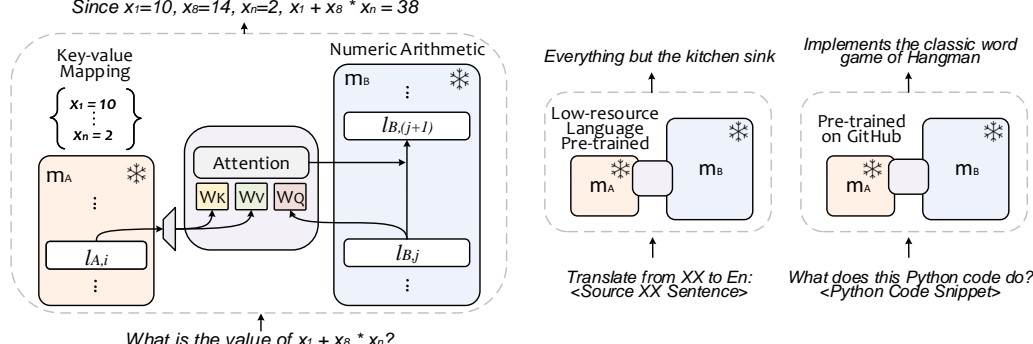

Figure 1: **Overview of CALM.** To augment an *anchor* LLM ($\mathbf{m_B}$) with new capabilities through *composition* with a specialized *augmenting* model ($\mathbf{m_A}$). Figure illustrates three $\mathbf{m_A}$ with different capabilities: key-value mapping (*left*), low-resource languages (*center*), and code (*right*). Models $\mathbf{m_A}$ and $\mathbf{m_B}$ remain unchanged (❄) during composition. A few additional parameters are learnt over models' layer representations. Leftmost plot shows an $\mathbf{m_A}$ trained on a set of string-integer mappings, e.g., $\{x_1 : 10, \ldots, x_n : 2\}$. $\mathbf{m_B}$ is a large LM with arithmetic capabilities. CALM composes these two frozen models to solve the task of arithmetic on keys which either models could not solve on their own (§4.1). Notably, CALM generalizes to the entire key-value set despite training with arithmetic examples spanning only 20% of the keys.

To address the training and the data challenges mentioned above, we propose and study a practical setting for *model composition*: (i) we are given access to one (or more) augmenting model(s) and an anchor model, (ii) we are *not allowed* to modify the weights of either models, and (iii) we only have access to a small amount of data, representing the "combined skills" of the given models, e.g., code generation with complex logical reasoning.

Prior work has largely approached the question of composition from either a routing or a merging standpoint, neither of which provide an effective solution to capture this setting. Routing between the given models, i.e., choosing an output of one model over the other (Ma et al., 2019), or performing a soft ensemble (Muqeeth et al., 2023) is not effective when neither of the models can demonstrate the desired capability. Another body of work creates a combined model by an arithmetic combination of base model parameters (Wortsman et al., 2022; Ilharco et al., 2022; Matena & Raffel, 2022). However, these settings are naturally restrictive and their efficacy is unclear when combining models with different sizes and pre-training objectives (Yadav et al., 2023).

In this work, we propose a novel **Composition to Augment Language Models** (**CALM**) framework to address the general model composition setting mentioned above. Rather than a shallow combination of the augmenting and anchor LMs (Wortsman et al., 2022; Ilharco et al., 2022), CALM introduces a small number of trainable parameters over both augmenting and anchor models' intermediate layer representations. CALM finds an effective combination of the given models to perform new challenging tasks more accurately than either of the models alone, while preserving the capabilities of individual models. Figure 1 highlights few motivating scenarios for CALM.

We study key practical applications of CALM: language inclusivity and code generation. For language inclusivity (§4.2), we use a model that has been trained on a set of low-resource languages. We observe that composing this model with the LLM allows us to borrow its generation and reasoning capabilities to achieve significantly better performance on translation and arithmetic reasoning tasks for low-resource languages (Tables 2 and 3). This composed model outperforms not only the two base models but also versions of the LLM that have been further pre-trained or LoRA (Hu et al., 2021) fine-tuned for the set of low-resource languages. For code generation (§4.3), we use a model that has been trained on open-source code across a variety of programming languages. Composing this model with the LLM—hence borrowing its low-level logic and generation capabilities—outperforms the two base models (Table 4) on code explanation and code completion tasks.

## 2 RELATED WORKS

**Parameter efficient fine-tuning:** A large body of prior work focuses on parameter efficient ways of fine-tuning models for new tasks by introducing a small number of trainable parameters, keeping the original model intact (Houlsby et al., 2019; Wang et al., 2020; Pfeiffer et al., 2021; Hu et al., 2021; Kessler et al., 2022). Since this paradigm allows a small set of new parameters to be trained, it is challenging to use these approaches to augment novel domains and knowledge sources that are entirely absent from the original training corpus. In contrast, CALM enables a model to be adapted to new domains using augmenting models. In Section 4.4, we draw empirical comparisons between CALM and LoRA (Hu et al., 2021), a representative parameter efficient fine-tuning method.

**Model Merging:** Merging different expert models with simple techniques like task vector averaging provides a way of recombining different capabilities of these models (Ilharco et al., 2022; Matena & Raffel, 2022). However, these methods are only relevant when the original models are well aligned. Other related approaches are also applicable only when the models are derived from the same model (Matena & Raffel, 2022) or they are of same size (Muqeeth et al., 2023). In contrast, CALM is more generic and is applicable to any set of models.

**Model and Task Compositionality:** The modular encoder-decoder based method in Dalmia et al. (2022) adapts components of encoder-decoder models to allow flexible re-usability of different encoders, each with their own capabilities. Several past studies explore compositionality of modality-specific encoders with language models to serve multi-modal use-cases (Ziegler et al., 2019; Alayrac et al., 2022). Typically, they introduce cross-attention parameters across a language model in order to attend to representations from an image encoder and show an effective transfer of modalities across models. In this work, we extend the ideology of model re-use and modularity to composition of capabilities in large language models.

**Models as Tools:** Another interesting direction for using multiple language models to solve a downstream task has been to perform composition in the models' input text space (Zeng et al., 2022; Shen et al., 2023). Schick et al. (2023) have demonstrated how a model can be taught to use external tools—there might be an opportunity to investigate if other models can be called as a part of the same framework. Since these approaches require a large amount of prompt engineering, in this work we focus on composition through representations that can be learnt automatically.

## 3 COMPOSITION TO AUGMENT LANGUAGE MODELS (CALM)

Given an ***anchor model*** $\mathbf{m_B}$ and an ***augmenting model*** $\mathbf{m_A}$, CALM aims to compose the two models ($\mathbf{m_{A \oplus B}}$) to enable new capabilities as a composition of capabilities of the two individual models.

As discussed in the introduction, we study this composition in a practical setting with the following assumptions: i) we can access weights, run forward and backward pass, and access intermediate representations of both $\mathbf{m_B}$ and $\mathbf{m_A}$, ii) we are not allowed to change weights of both the models, iii) we do not have access to the training data, hyperparameters, training states of both the base models, iv) we are provided a few examples from the target composition domain.

The goal is to learn a composition $\mathbf{m_{A \oplus B}} = f(\mathbf{m_A}, \mathbf{m_B}, \Theta_\mathbf{C}, \mathbf{D_C})$ to achieve some joint task C. The weights of $\mathbf{m_A}$ and $\mathbf{m_B}$ are frozen. $\Theta_\mathbf{C}$ is the additional set of trainable parameters introduced to learn the composition and $\mathbf{D_C}$ refers to the set of examples that are used to learn this composition.

### 3.1 LEARNING TO COMPOSE ($\Theta_\mathbf{C}$)

As outlined in Figure 1, we operate over a selected set of layers from $\mathbf{m_B}$ and $\mathbf{m_A}$ at all times. We learn two sets of additional parameters over these layers: (i) A simple set of linear transformations, $f_{\text{proj}}(.)$ that maps an $i^{\text{th}}$ layer representation from $\mathbf{m_A}$ to the dimensionality of representations from $\mathbf{m_B}$, and (ii) A set of cross-attention layers, $f_{\text{cross}}(.,.)$ that cross-attend between this transformed layer representation and a $j^{\text{th}}$ layer representation from $\mathbf{m_B}$.

**Compositional Layers:** Let the augmenting model $\mathbf{m_A}$ and the anchor model $\mathbf{m_B}$ have $N_A$ and $N_B$ layers, respectively. Also, let $D_A$ and $D_B$ be the token dimensionality of the two models. We

first choose a set of *compositional* layers—$\mathbb{L}_A$ and $\mathbb{L}_B$—for both models, over which the set of new learnable parameters are introduced during composition. $n_A = |\mathbb{L}_A|$ and $n_B = |\mathbb{L}_B|$. For simplicity, we set $n_A = n_B = n$ and the gap between two contiguous selected layers is kept uniform based on the number of selected layers—that is, $(l_2 - l_1) = \cdots = (l_n - l_{(n-1)}) = N/n$. Further, $\mathbb{H}_A \in \{\mathbf{H}_{A1}, \mathbf{H}_{A2}, \ldots, \mathbf{H}_{An_A}\}$ denote the layer representation of a given input after each layer in $\mathbb{L}_A$.

**Learned Projections:** Next we map representations from $\mathbf{m}_A$ to that of $\mathbf{m}_B$ via a projection layer. In particular, for each layer in $\mathbb{L}_A$, we learn a projection function $f_{\text{proj}} : \mathbb{R}^{D_A} \to \mathbb{R}^{D_B}$, that projects representations from these layers to the desired representation size of $\mathbf{m}_B$. Let,

$$f_{\text{proj}}(\mathbb{H}_A) \leftarrow \{f_{\text{proj}}(\mathbf{H}_{A1}), f_{\text{proj}}(\mathbf{H}_{A2}), \ldots, f_{\text{proj}}(\mathbf{H}_{An_A})\}$$

This transformation enables cross-attention across models, and also performs an alignment of representations from $\mathbf{m}_A$ and $\mathbf{m}_B$ despite frozen weights of the base models.

**Cross-attention Layers:** Similar to the multi-headed cross-attention in encoder-decoder models (for example Vaswani et al. (2017) and Raffel et al. (2020))—we introduce cross-attention between representations of the anchor and the augmenting model. In particular, we use $f_{\text{proj}}(\mathbf{H}_{Ai})$ from the augmenting model as the *key* and *value* vectors for each head in cross-attention. We use the vector $\mathbf{H}_{Bj}$ from the anchor model as the *query* vector, which leads to the following cross-attention setup:

$$f_{\text{cross}}(f_{\text{proj}}(\mathbf{H}_{Ai}), \mathbf{H}_{Bj}) = \text{Concat.}_k\, (\text{head}_k)\, \mathbf{W}^O \quad \forall k \in N_H$$
$$\text{where, head}_k = \text{Attn.}(\mathbf{Q}_B, \mathbf{K}_A, \mathbf{V}_A),$$
$$\text{and, } \mathbf{Q}_B = \mathbf{H}_{Bj}\mathbf{W}_k^Q,$$
$$\mathbf{K}_A, \mathbf{V}_A = f_{\text{proj}}(\mathbf{H}_{Ai})\mathbf{W}_k^K,\; f_{\text{proj}}(\mathbf{H}_{Ai})\mathbf{W}_k^V$$

Here, $N_H$ represents the number of attention heads used for cross-attention which, in our case, is typically the same as the number of heads used for self-attention in $\mathbf{m}_B$. Each of $\mathbf{W}^O \in \mathbb{R}^{D_B \times D_B}$, $\mathbf{W}_k^Q, \mathbf{W}_k^K,$ and $\mathbf{W}_k^V \in \mathbb{R}^{D_B \times D_B // N_H}$ are learnable weight matrices, where $k \in \{1..N_H\}$.

Finally, the cross-attention output is added as a residual connection to the layer representations of $\mathbf{m}_B$. The resultant output vector, in-turn, is the input to the succeeding layer in $\mathbf{m}_B$:

$$\mathbf{H}_{A \oplus Bj} = \mathbf{H}_{Bj} + f_{\text{cross}}(f_{\text{proj}}(\mathbf{H}_{Ai}), \mathbf{H}_{Bj})$$

Here, $\mathbf{H}_{A \oplus Bj}$ denotes the input to the $(j+1)^{th}$ layer of the composed model. All layers in $\mathbb{L}_A$ and $\mathbb{L}_B$ are utilized in a similar manner. Propagating over the remaining layers in $\mathbf{m}_B$ gives us a final output token $y_t$ decoded for the $t^{th}$ timestep. Akin to usual auto-regressive decoding, the output token for each time-step is appended to the input: $x_{t+1} = x_t \oplus y_t$, Since the updated input at each time step is passed to both models, all representations for the two models are refreshed.

## 3.2 Composition Training Data ($\mathbf{D_C}$)

Since the target model $\mathbf{m}_{A \oplus B}$ involves a composition over the two models $\mathbf{m}_A$ and $\mathbf{m}_B$, we construct the set of training examples $\mathbf{D_C}$ to depict a "combined skill" that enables $\Theta_C$ to attend over the two models appropriately for the target task.

Ideally, if the set of tasks involved in composition task are distinguished as $\mathbf{t}_1$ and $\mathbf{t}_2$ respectively, then we design $\mathbf{D_C}$ to depict a joint task $\mathbf{C}$. For example, with respect to our synthetic key-value setup: our final task ($\mathbf{C}$) is to perform arithmetic over a set of keys. The augmenting model $\mathbf{m}_A$ is trained to learn the given key-value pairs (notated as task, $\mathbf{t}_1$) and the anchor model $\mathbf{m}_B$ is generic model that can perform numeric arithmetic well (task $\mathbf{t}_2$). For learning the set of parameters $\Theta_C$ for composition, we consider $\mathbf{D_C}$ to be arithmetic over a held-in set of keys (task $\mathbf{C}$), encompassing combined skills from the two models. In contrast to fine-tuning approaches like LoRA (Hu et al., 2021) that would require the entire knowledge source (here, key-values) during training time, we find that training composition on only a fraction of the keys can generalize to the full set.

In other real world settings, a clear distinction in specializing tasks for each model might be difficult to formulate and hence defining a task that captures the combined skills can be challenging. We find

that using a set of examples that capture certain capabilities of the two models suffices, i.e., some rough notion of $\mathbf{t}_{A \cup B}$. For our language inclusivity task, we use a mixture of examples containing a small amount of low-resource language and high-resource language data.

**Composing multiple models:** Finally, we note that while the method has been presented for a setting with one anchor model and only one augmenting model, CALM is applicable to multiple augmenting models as well. In particular, CALM would require learning similar projection and cross-attention components between the anchor and each of the augmenting model. We leave a thorough investigation of this as a topic of future work.

## 4 EXPERIMENTS

We demonstrate the following in three domains: **(a)** an anchor LLM ($\mathbf{m}_B$) can be composed with an augmenting model ($\mathbf{m}_A$) trained on mappings between string keys and number values to solve arithmetic expressions over those keys requiring both, knowledge of the KV mappings and arithmetic capabilities (§4.1); **(b)** how CALM can be used to expand the language coverage of an anchor LLM ($\mathbf{m}_B$) to low-resource languages it has not seen during pre-training. We show that an augmenting model ($\mathbf{m}_A$) pre-trained on low-resource languages can be composed with such an anchor model to significantly improve translation and math-word problem solving capabilities in low-resource languages (§4.2); **(c)** how code completion and explanation can be improved by composing an anchor LLM with an augmenting model ($\mathbf{m}_A$) specializing in the code domain (§4.3).

In all experiments, we start with a PaLM2-XXS model and further train it on domain-specific data to arrive at an augmenting model ($\mathbf{m}_A$) that is then kept frozen during composition. Note that no task specific training data was used to train CALM. We use PaLM2-XS or PaLM2-S models as the anchor LLM ($\mathbf{m}_B$) that is also kept frozen during composition training. For all our experiments, we set $N_A/n = 4$, i.e., we perform composition using every 4th layer output from $\mathbf{m}_A$. Correspondingly, layers from $\mathbf{m}_B$ ($\mathbb{L}_B$) are chosen such that $n_B = n_A = n$, hence $n_B = N_B/4$.

### 4.1 KEY-VALUE ARITHMETIC

We first study the setting where we have a small augmenting LM that has been trained to memorize string-to-integer key-value (KV) mappings, and a large anchor LM that is capable of performing arithmetic over integers. We wish to use CALM to compose them and enable a new capability of solving arithmetic expressions containing those keys.

**Key-Value Domain Knowledge** We first generate a repository of KV pairs containing $N_{KV} = 25K$ pairs by sampling English strings of $2-6$ characters from the vocabulary of the PaLM2-XXS model and randomly assigning them unique integer values in the range $[1, N_{KV}]$. This constitutes the knowledge artifact, $\mathbf{D}_{KV}$. We further generate a collection of arithmetic expressions ($\mathbf{D}_{KV\text{-}EXP}$) containing addition ($+$), subtraction ($-$), and multiplication ($\times$) operations between $3-6$ keys by randomly sampling keys from $\mathbf{D}_{KV}$ and operations to perform between them. We generate three datasets:

**(i)** KV-Substitution ($\mathbf{D}_{KV\text{-}SUBS}$): This dataset maps each expression in $\mathbf{D}_{KV\text{-}EXP}$, to an expression where the keys are replaced by their corresponding values. For example, this dataset contains examples of the form $(\texttt{<K1>} + \texttt{<K2>} - \texttt{<K3>}, 10 + 22 - 24)$.

**(ii)** KV-Arithmetic ($\mathbf{D}_{KV\text{-}MATH}$): This dataset maps each expression in $\mathbf{D}_{KV\text{-}EXP}$ to the numeric value arrived at by solving the arithmetic expression when the keys would be replaced by the corresponding values. For example, examples in this dataset look like $(\texttt{<K1>} + \texttt{<K2>} - \texttt{<K3>}, 8)$.

**(iii)** Numeric-Arithmetic ($\mathbf{D}_{NUM\text{-}MATH}$): This dataset maps the value substituted version of each expression in $\mathbf{D}_{KV\text{-}EXP}$ to the numeric value arrived at by solving the arithmetic expression. For example, examples in this dataset look like $(10 + 22 - 24, 8)$.

**Models** We obtain augmenting model $\mathbf{m}_A$ by further training a pre-trained PaLM2-XXS model on $\mathbf{D}_{KV\text{-}SUBS}$ to make it memorize the KV pairs in $\mathbf{D}_{KV}$. Note that, training on $\mathbf{D}_{KV\text{-}SUBS}$ does not teach this augmenting model how to solve arithmetic expressions. Next, we use a pre-trained PaLM2-XS model as the anchor model $\mathbf{m}_B$. This model is capable of solving numeric expressions with decent performance (see Table 1). Note that, this model has no knowledge of the KV pairs in $\mathbf{D}_{KV}$.

We now take examples from the KV-Substitution dataset $\mathbf{D}_{\text{KV-SUBS}}$ that only span $20\%$ of the keys in $\mathbf{D}_{\text{KV}}$ to form the training data for composition ($\mathbf{D}_{\text{C}}$). We use $\mathbf{D}_{\text{C}}$ to compose the augmenting model ($\mathbf{m}_{\text{A}}$) having knowledge of $\mathbf{D}_{\text{KV}}$ and the pre-trained anchor model $\mathbf{m}_{\text{B}}$ by training the composition parameters ($\Theta_{\text{C}}$) using CALM as explained in §3. Both $\mathbf{m}_{\text{A}}$ and $\mathbf{m}_{\text{B}}$ are kept unchanged.

**Evaluation Task**   We evaluate the composed model $\mathbf{m}_{\text{A}\oplus\text{B}}$ for its ability to solve arithmetic expressions containing keys from $\mathbf{D}_{\text{KV}}$. Specifically, we evaluate on the subset of $\mathbf{D}_{\text{KV-MATH}}$ dataset that does not contain expressions used in $\mathbf{D}_{\text{C}}$ during training. This way, we are able to measure the composed model's ability to generalize to keys beyond what was observed during training.

**Results**   Table 1 shows the performance of the three models: $\mathbf{m}_{\text{A}}$, $\mathbf{m}_{\text{B}}$, and $\mathbf{m}_{\text{A}\oplus\text{B}}$ across the aforementioned datasets. First, we observe that the augmenting model $\mathbf{m}_{\text{A}}$ achieves $98.1\%$ at the KV-Substitution task showing that memorizes $\mathbf{D}_{\text{KV}}$ well. Next, we see that it performs poorly ($4.2\%$) at the Numeric-Arithmetic task showing that it does not have arithmetic capabilities. As a result, this model is not able to solve arithmetic expressions containing keys from $\mathbf{D}_{\text{KV}}$.

As expected, the anchor model $\mathbf{m}_{\text{B}}$ gets $0\%$ accuracy on the KV-Substitution and KV-Arithmetic tasks as it has not seen any data from $\mathbf{D}_{\text{KV}}$. However, it performs well ($73.7\%$) on the Numeric-Arithmetic task demonstrating capability of arithmetic over numerals.

Lastly, we see that the composed model $\mathbf{m}_{\text{A}\oplus\text{B}}$ is able to solve all tasks with high accuracy, especially the KV-Arithmetic task ($84.3\%$) for which both the underlying

|  | $\mathbf{m}_{\text{A}}$ | $\mathbf{m}_{\text{B}}$ | CALM ($\mathbf{m}_{\text{A}\oplus\text{B}}$) |
|---|---|---|---|
| $\mathbf{D}_{\text{KV-SUBS}}$ | 98.1 | 0.0 | 92.9 |
| $\mathbf{D}_{\text{NUM-MATH}}$ | 4.2 | 73.7 | 72.0 |
| $\mathbf{D}_{\text{KV-MATH}}$ | 0.7 | 0.0 | **84.3** |

Table 1: Evaluation (accuracy (%)) for a synthetic key-value (KV) task.   $\mathbf{m}_{\text{A}}$ is trained to memorize the KV mappings while $\mathbf{m}_{\text{B}}$ excels at arithmetic We see that a composition $\mathbf{m}_{\text{A}\oplus\text{B}}$ is able to perform arithmetic over held-out keys.

models fail. This shows that the composed model is able to leverage the relevant capabilities from both the augmenting and anchor model to solve a complex task.

### 4.2   Low-resource Language Inclusivity

| Model | FLORES-200 (XX to En; chrF1) | | | | | | | | | | |
|---|---|---|---|---|---|---|---|---|---|---|---|
| | lij | mr | taq | nn | su | ban | pl | th | min | acm | *avg.* |
| PaLM2-XXS | 24.0 | 16.5 | 21.6 | 33.3 | 20.6 | 2.1 | 5.3 | 63.2 | 44.0 | 59.8 | 29.0 |
| + NTL ($\mathbf{m}_{\text{A}}$) | 32.0 | 21.6 | 46.9 | 50.0 | 40.6 | 4.1 | 4.0 | 63.8 | 47.8 | 61.1 | 37.2 |
| PaLM2-S ($\mathbf{m}_{\text{B}}$) | 32.6 | 24.2 | 44.6 | 50.8 | 50.9 | 5.4 | 9.5 | 69.0 | 61.0 | 68.6 | 41.7 |
| CALM ($\mathbf{m}_{\text{A}\oplus\text{B}}$) | **44.1** | **30.4** | **55.1** | **54.6** | **54.4** | **11.8** | **11.3** | **69.4** | **61.1** | **68.9** | **46.1** |
| $\mathbf{m}_{\text{B}}$+NTL ($\mathbf{m}_{\text{B}}^{\text{NTL}}$) | 48.1 | 39.1 | 59.2 | 57.5 | 57.3 | 11.4 | 9.9 | 69.4 | 61.4 | 69.0 | 48.2 |

Table 2: Translation performance for XX to English direction on the FLORES-200 dataset (Costa-jussà et al., 2022): We show results for a subset of 10 low-resource languages. Note that the composed model $\mathbf{m}_{\text{A}\oplus\text{B}}$ significantly outperforms both $\mathbf{m}_{\text{A}}$ and $\mathbf{m}_{\text{B}}$. On the complete language list, $\mathbf{m}_{\text{A}\oplus\text{B}}$ outperforms both the underlying models for 175 of 192 languages (Appendix A; Figure 2). $\mathbf{m}_{\text{B}}^{\text{NTL}}$ represents a skyline where $\mathbf{m}_{\text{B}}$ has been further pre-trained on $\mathbf{D}_{\text{NTL}}$. The composed model achieves similar performance for a tiny fraction of the training cost.

In this section, we study if we can compose such a large anchor LM $\mathbf{m}_{\text{B}}$ with a smaller augmenting LM $\mathbf{m}_{\text{A}}$ that has been pre-trained on low-resource languages, to perform translation and math-word problem solving tasks presented in these low-resource languages.

**Low-resource Language Corpora**   We use the long-tail language set and the associated corpora from the Next Thousand Languages (NTL) effort (Caswell et al., 2020; Bapna et al., 2022) as the

| Model | GSM8K (Low-resource Languages; Accuracy) | | | | | | | | | | |
|---|---|---|---|---|---|---|---|---|---|---|---|
| | meo | mfa | pcm | efi | min | ilo | ady | mai | nso | mzn | *avg.* |
| PaLM2-XXS | 5.2 | 6.8 | 6.8 | 4.0 | 5.6 | 7.2 | 6.0 | 3.6 | 7.2 | 6.8 | 5.9 |
| + NTL ($m_A$) | 7.6 | 4.0 | 4.4 | 3.2 | 6.0 | 4.8 | 6.4 | 3.2 | 6.0 | 4.8 | 5.0 |
| PaLM2-S ($m_B$) | 28.8 | 14.0 | **34.4** | 14.8 | **25.2** | 14.8 | 30.0 | 22.8 | 8.4 | 31.6 | 22.5 |
| CALM ($m_{A\oplus B}$) | **34.0** | 17.6 | 33.6 | **18.0** | 23.6 | **16.8** | **36.4** | **24.8** | 8.4 | **36.4** | **25.0** |
| $m_B^{NTL}$ | 33.2 | 20.4 | 31.6 | 14.0 | 24.8 | 14.0 | 29.2 | 21.2 | 9.6 | 27.6 | 22.6 |

| Model | (High-resource Languages) | | | | | | | | | | |
|---|---|---|---|---|---|---|---|---|---|---|---|
| | en | te | bn | sw | ja | zh | th | fr | es | de | *avg.* |
| PaLM2-XXS | 5.6 | 4.0 | 2.0 | 7.6 | 2.0 | 4.4 | 6.0 | 6.8 | 5.6 | 9.2 | 5.3 |
| + NTL ($m_A$) | 4.8 | 3.6 | 3.2 | 4.8 | **3.2** | 7.6 | 6.4 | 9.2 | 5.6 | 7.2 | 5.6 |
| PaLM2-S ($m_B$) | 36.8 | 19.2 | 23.2 | 16.0 | 2.0 | 39.2 | 29.6 | 38.0 | 32.4 | 43.2 | 28.0 |
| CALM ($m_{A\oplus B}$) | **37.2** | **28.0** | **27.2** | **18.0** | 2.4 | **43.6** | **33.2** | **42.8** | **36.0** | **49.2** | **31.8** |
| $m_B^{NTL}$ | 36.0 | 17.6 | 18.4 | 14.4 | 0.8 | 33.6 | 27.2 | 34.8 | 31.2 | 42.0 | 25.6 |

Table 3: Evaluations for grade-school mathematics (GSM) problems on low-resource (LRL) and high-resource (HRL) languages. We observe that CALM yields significant gains for both evaluation sets. Gains on the HRL set suggests that CALM avoids catastrophic forgetting.

domain data $D_{NTL}$. This large-scale corpora contains web-crawled monolingual sentences and translation pairs for $\sim$1000 languages. The dataset has been used for language expansion in translation systems and language models (Garcia et al., 2021; Siddhant et al., 2022).

**Models**   Akin to §4.1, we obtain augmenting model $m_A$ by training the PaLM2-XXS model on $D_{NTL}$ to impart knowledge about these low-resource languages to the model. For $m_B$, we use the pre-trained PaLM2-S model. We use $\sim 5\%$ of the same low-resource language corpora $D_{NTL}$ as the training data $D_C$ to compose $m_A$ and $m_B$ via CALM. Since both models are untrained during composition, the anchor model $m_B$ is *not* trained on any of the low-resource language data.

**Evaluation Tasks**   We evaluate the composed model $m_{A\oplus B}$ on two tasks:

**(i)** Translating text from a non-English language to English: We carry out these evaluations in a 5-shot in-context learning paradigm on the FLORES-200 (Costa-jussà et al., 2022) dataset. This dataset contains examples for 200 high- and low-resource languages.

**(ii)** Performing grade school math word problems expressed in a non-English language: We evaluate on the multilingual version of the GSM-8K dataset (Shi et al., 2023) containing math word problems for English and 9 other high-resource languages. We further generated a silver-standard GSM-8K dataset for low-resource languages by automatically translating the English examples in GSM-8K to 25 low-resource languages supported by Google Translate.[1]

**Results**   Table 2 shows results on the FLORES-200 dataset (Costa-jussà et al., 2022), where the input is a low-resource (XX) language sentence and the output should be the corresponding English translation. For 10 low-resource languages shown in the Table, we see that both the underlying models $m_A$ and $m_B$ are outperformed by our composed model $m_{A\oplus B}$. We find that the composed model $m_{A\oplus B}$ outperforms $m_B$ on 175 of the complete set of 192 languages (Appendix A).

Table 3 shows the performance of these models on the grade-school math word problems from the GSM8K task (Cobbe et al., 2021) on low-resource languages (*top*) and high-resource languages (Shi et al. (2023); *bottom*). Firstly, we observe that the augmenting model $m_A$ does not perform well on this task due to its limited mathematical reasoning capabilities. On the other hand, the anchor model $m_B$ does much better given its mathematical reasoning capabilities and transfer-learning from high-resource languages. Finally, we observe that $m_{A\oplus B}$ outperforms both $m_A$ and $m_B$ on **18 of 25** low-resource and **9 of 10** high-resource languages, demonstrating effective composition of models.

---

[1] We perform quality evaluations in Appendix 7.

| Model | CC (P@1) | T2C (P@1) | C2T (chrF1) | | | | | |
| | HumanEval | MBPP | Python | PHP | Go | Java | JS | Ruby |
|---|---|---|---|---|---|---|---|---|
| PaLM2-XXS + Code ($\mathbf{m_A}$) | 19.5 | 28.0 | 28.0 | 34.7 | 32.6 | 29.6 | 26.5 | 26.0 |
| PaLM2-S ($\mathbf{m_B}$) | 16.4 | 28.6 | 30.4 | 35.5 | 40.4 | 31.0 | 28.8 | 27.9 |
| CALM ($\mathbf{m_{A\oplus B}}$) | **22.5** | **32.2** | **30.5** | **35.8** | **40.6** | **31.4** | **29.3** | **29.0** |
| $\mathbf{m_B^{Code}}$ | 24.3 | 43.0 | 18.9 | 35.0 | 41.1 | 31.1 | 20.2 | 27.6 |

Table 4: Evaluations for code generation and understanding across three tasks: Code Completion (CC), Text-to-Code (T2C), and Code-to-Text (C2T). Augmenting code understanding to $\mathbf{m_B}$ using $\mathbf{m_A}$ significantly improves performances across all datasets. $\mathbf{m_B^{Code}}$ represents a skyline where $\mathbf{m_B}$ further pretrained on the $\mathbf{D_{Code}}$, which shows catastrophic forgetting of text generation task.

See Table 6 (Appendix A.2) for a complete set of evaluations. Note that the last row in Table 3 shows that $\mathbf{m_B}$ when fine-tuned on $\mathbf{D_{NTL}}$ leads to worse performance than the pre-trained $\mathbf{m_B}$ indicating forgetting. Composing domain-specific model $\mathbf{m_A}$ with $\mathbf{m_B}$ using CALM avoids this.

## 4.3 CODE UNDERSTANDING AND GENERATION

Code understanding and generation require two distinct types of capabilities: (**a**) knowledge of the syntax and semantics of code, and (**b**) knowledge of the world that the code is manipulating. While LLMs have a wealth of world knowledge, they could often lack the specific knowledge of code syntax due to a skewed representation of code data in their pretraining corpora. Conversely, small models trained specifically on code data could exhibit a good understanding of code syntax, but they may lack broad world knowledge and reasoning. CALM can enable best of both worlds.

**Code Domain Data** Here, we use the code-specific corpus, $\mathbf{D_{Code}}$, consisting of open-source code extracted from GitHub heads for a variety of programming languages to train $\mathbf{m_A}$.

**Models** Similar to §4.1, a version of the PaLM2-XXS model has been further pre-trained on $\mathbf{D_{Code}}$ is used as $\mathbf{m_A}$, while the base pre-trained PaLM2-S model acts as $\mathbf{m_B}$. We build $\mathbf{m_{A\oplus B}}$ by training CALM with only 7% of the same code data (data used for $\mathbf{m_A}$) to have a data parity.

**Evaluation Tasks** We evaluate the efficacy of CALM on three different tasks:

(**i**) Code-Completion (CC): Given an initial set of lines of a code, the model is prompted to complete the code snippet. Here the aim is to evaluate the model for code syntax. We perform zero-shot evaluations on HumanEval benchmark dataset (Chen et al., 2021) and report the Pass@1 (P@1) metric.

(**ii**) Text-to-Code (T2C): Given a textual context, the model is prompted to generate the corresponding code snippet. Here, the evaluation indicates language understanding and code generation capabilities. We perform 3-shot inference on the MBPP dataset (Austin et al., 2021) and report P@1.

(**iii**) Code-to-Text (C2T): Given a code snippet, the goal is to generate a natural language explanation of the code. This task evaluates code understanding and text generation. We perform 3-shot evaluations on the CodeXGlue benchmark (Lu et al., 2021) and report chrF1 scores across languages.

**Results** Table 4 reports comparative performance for the individual models $\mathbf{m_A}$ and $\mathbf{m_B}$, the composed version $\mathbf{m_{A\oplus B}}$, and a fine-tuned anchor baseline $\mathbf{m_B^{Code}}$. Firstly, evaluations on the HumanEval dataset suggest that $\mathbf{m_A}$ has a superior understanding of code syntax as a result of its additional training on $\mathbf{D_{Code}}$. While, due to the larger scale and general purpose pre-training of $\mathbf{m_B}$, it excels at general language understanding and hence performs better on the T2C and C2T tasks.

When employing CALM to compose the two models, we observe a clear transfer and composition of capabilities through significant performance improvements: 6.1% and 3.6% absolute gains over $\mathbf{m_B}$ on the CC and T2C tasks, respectively. We observe that fine-tuning $\mathbf{m_B}$ on $\mathbf{D_{Code}}$ leads to a significant decline in the C2T performance due to catastrophic forgetting. CALM retains the perfor-

| | FLORES-200 (XX-En) | | GSM-8K (LRL) | | GSM-8K (HRL) | | CC | T2C | C2T |
|---|---|---|---|---|---|---|---|---|---|
| | chrF1 | #($>m_B$) | acc. | #($>m_B$) | acc. | #($>m_B$) | P@1 | P@1 | chrF1 |
| $m_{A⊕B}$ | **60.5** | **175** | **21.4** | **20** | **33.1** | **11** | **22.5** | **32.2** | **32.6** |
| Vanilla $m_A$ | 59.2 | 115 | 19.0 | 15 | 29.7 | 8 | 20.0 | 28.0 | 32.2 |
| Random $m_A$ | 58.8 | 43 | 17.8 | 9 | 28.5 | 4 | 20.1 | 27.0 | 32.1 |
| $m_A$ as an encoder | 59.3 | 102 | 19.1 | 12 | 29.1 | 6 | 16.0 | 27.0 | 32.0 |
| LoRA | 59.2 | 82 | 20.9 | 15 | 31.2 | 9 | 18.3 | 28.7 | **32.6** |

Table 5: Comparative performance of CALM ($m_{A⊕B}$) across various possible ablations. The metric "#($>m_B$)" depicts the number of languages for which the corresponding model is better than the base for NTL, $m_B$—out of 192, 25, and 11 languages for the three tasks respectively.

mance and is marginally better than $m_B$ across all languages. We also study qualitative examples on the C2T task and observe interesting common patterns that are discussed in Appendix B.

### 4.4 ABLATIONS

**Influence of $m_A$** We first study the influence of $m_A$ by replacing it with vanilla and random variants during composition. Table 5 shows the variation of performance across NTL and Code tasks when the specialized $m_A$ is replaced with a *vanilla* PaLM2-XXS checkpoint or an untrained version of the model, i.e., a *random* model. We see that there is a considerable drop of performance with these variants across all tasks. On FLORES-200 XX-En task, languages improved with composition drop to 115 and 43 with vanilla and random, respectively. A slight improvement of the vanilla model over $m_B$ indicates that an un-specialized model (with a different training regime than $m_B$) might have orthogonal capabilities leading to an enhanced model. This finding validates that performance gains seen with CALM is a result of utilizing $m_A$ and not just the additional $Θ_C$ parameters.

**Influence of iterative decoding** We also investigate a variation where we use $m_A$ as an encoder, i.e., an output token decoded at a given timestep is not amended to $m_A$'s input. In this case, only the prefix representations of $m_A$ are used. This setting eludes to past work for image and text models (Alayrac et al., 2022) where encoder and decoder models are composed. We observe a significant decline in performance across our various tasks when employing this setting.

**Comparision with LoRA** Finally, we evaluate a parameter efficient fine-tuning approach by training LoRA (Hu et al., 2021) layers to adapt $m_B$. For all experiments, we set the LoRA rank such that the number of added parameters is equal to the number of parameters introduced with CALM. We also train LoRA on the same data as CALM, i.e., $D_C$. We see a considerable difference in performance between the two approaches across all tasks and metrics.

## 5 CONCLUSION

The proposed CALM framework composes an *anchor* LLM with specialized *augmenting* models to enable new tasks not achievable by either models individually. CALM does not require updating the individual models and learns a dense interaction between the models through a few trainable cross-attention parameters. Our experiments present consistent evidence that CALM learns to utilize the expertise from the two models. That is, when composed with relevant augmenting models, we observe a significant uptick in the anchor model's performance across multiple challenging tasks, such as low-resource translation, reasoning, and code explanation/generation.

That is, CALM is especially useful in scenarios where proprietary data and knowledge is stored in parametric models. With CALM, a foundational LLM could be augmented with such proprietary models to extend a variety of foundational capabilities such as reasoning, world knowledge, and coherent generation over the target proprietary domains. Finally, extensions of CALM could be used to acquire distinct knowledge from multiple augmenting models.

## Acknowledgments

This work was done during RB's pre-doctoral tenure at Google Research, India (GRI) with PT and PJ. RB is indebted to Manish Gupta, Divvy Thakkar, and all others who enabled this oppurtunity. RB would also like to thank the members of the Languages team and other researchers at GRI (and beyond), including the incredible pre-doctoral cohort. This work wouldn't have been possible without their constant support. Namely: Aishwarya P.S., Laurent El Shafey, and Qiao Zhang for their massive help in coding and debugging; Palak Jain and Sagar Gubbi for their feedback and support throughout the project; Kartikeya Badola, Shreyas Havaldar, Amandeep Kaur, and Rishabh Tiwari for being the first ears to all ideas; Cyrus Rashtchian and Richa Dixit for their mentorship.

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

# A    Supplementary Material for NTL

## A.1    FLORES-200

Figure 2 depicts the gains over the anchor PaLM2-S model when augmented with a model that has been trained on $\mathbf{D}_{\text{NTL}}$. We see a positive gain through CALM for **175 of 192** languages. The highest gains are seen for low-resource languages since they are the most underrepresented in the original model. Diminishing returns with higher resource languages is seen and this trend is similar to the trend seen for $\mathbf{m}_{\text{B}}^{\text{NTL}}$.

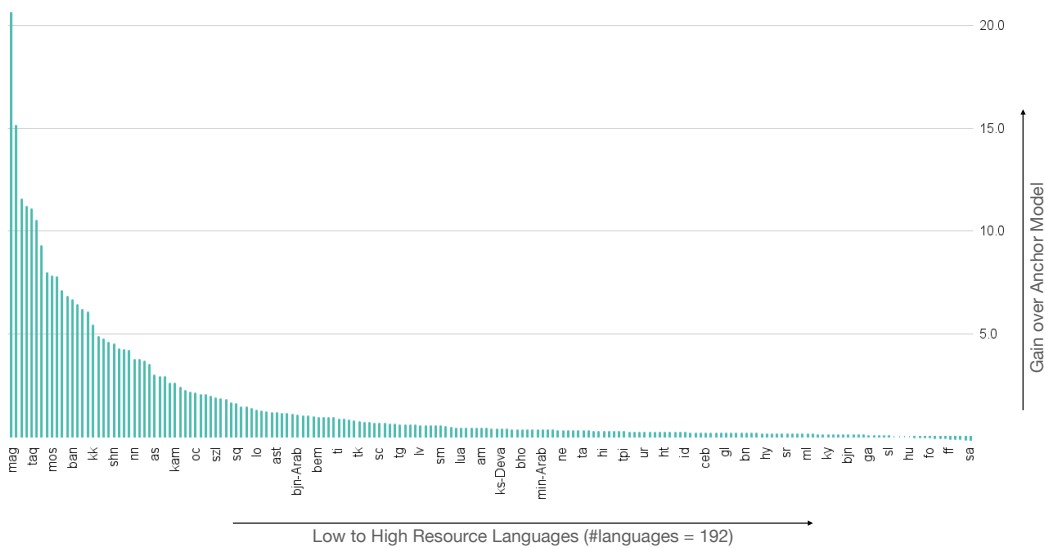

Figure 2: Gains seen by the composed model $\mathbf{m}_{\text{A}\oplus\text{B}}$ over the anchor model, $\mathbf{m}_{\text{B}}$, for the complete set of FLORES-200 languages. The languages are sorted from low to high-resource.

| | $\mathbf{m}_{\text{A}}$ | $\mathbf{m}_{\text{B}}$ | $\mathbf{m}_{\text{A}\oplus\text{B}}$ (CALM) | $\mathbf{m}_{\text{B}}^{\text{NTL}}$ | | $\mathbf{m}_{\text{A}}$ | $\mathbf{m}_{\text{B}}$ | $\mathbf{m}_{\text{A}\oplus\text{B}}$ (CALM) | $\mathbf{m}_{\text{B}}^{\text{NTL}}$ |
|---|---|---|---|---|---|---|---|---|---|
| meo | 7.6 | 28.8 | **34.0** | 33.2 | bho | 4.0 | 23.6 | **29.2** | 22.8 |
| mfa | 4.0 | 14.0 | **17.6** | 20.4 | cv | 6.0 | **17.6** | 16.4 | 20.4 |
| pcm | 4.4 | **34.4** | 33.6 | 31.6 | mni | 3.6 | 2.8 | **4.4** | 6.0 |
| efi | 3.2 | 14.8 | **18.0** | 14.0 | or | 2.4 | 9.6 | **12.4** | 12.0 |
| min | 6.0 | **25.2** | 23.6 | 24.8 | kri | 5.6 | 12.4 | **18.8** | 20.0 |
| ilo | 4.8 | 14.8 | **16.8** | 14.0 | tk | 5.2 | 27.2 | **29.2** | 28.8 |
| ady | 6.4 | 30.0 | **36.4** | 29.2 | gom | 4.8 | 22.4 | **25.2** | 22.8 |
| mai | 3.2 | 22.8 | **24.8** | 21.2 | ug | 6.0 | 23.2 | **29.2** | 26.4 |
| nso | 6.0 | **8.4** | **8.4** | 9.6 | ckb | 3.2 | 25.6 | **28.0** | 27.2 |
| mzn | 4.8 | 31.6 | **36.4** | 27.6 | as | 1.2 | 5.2 | **9.2** | 4.0 |
| bew | 4.4 | 33.6 | **34.8** | 33.6 | doi | 3.6 | 17.2 | **22.4** | 21.6 |
| ts | 4.8 | 7.2 | **10.0** | 11.6 | dz | **4.4** | 0.8 | 0.4 | 0.0 |
| dv | 2.8 | 11.2 | **14.8** | 13.2 | avg. | 4.5 | 18.6 | **21.4** | 19.8 |

Table 6: Performance evaluations on the complete set of low-resource languages for GSM-8K. Augmenting $\mathbf{m}_{\text{A}}$ with $\mathbf{m}_{\text{B}}$ as $\mathbf{m}_{\text{A}\oplus\text{B}}$ improves performance over $\mathbf{m}_{\text{B}}$ across a majority of languages. On average, we see an improvement of 2.8%.

|          | meo   | mfa   | pcm   | efi   | min   | ilo   | ady   |
|----------|-------|-------|-------|-------|-------|-------|-------|
| Overlap  | 83.17 | 75.54 | 81.28 | 78.35 | 77.90 | 77.80 | 76.21 |
| Delta    | 1.15  | 1.25  | 1.18  | 1.22  | 1.23  | 1.24  | 1.28  |
|          | mai   | nso   | mzn   | bew   | ts    | dv    | bho   |
| Overlap  | 76.63 | 69.58 | 71.32 | 71.37 | 61.62 | 55.18 | 73.67 |
| Delta    | 1.26  | 1.40  | 1.38  | 1.37  | 1.55  | 1.70  | 1.30  |
|          | cv    | mni   | or    | kri   | tk    | gom   | ug    |
| Overlap  | 58.52 | 58.94 | 68.03 | 77.18 | 66.06 | 71.21 | 57.66 |
| Delta    | 1.62  | 1.60  | 1.45  | 1.27  | 1.48  | 1.36  | 1.65  |

Table 7: Quality evaluation for the LRL GSM-8K dataset across languages. We created the dataset by translating the original English sentences of GSM-8K to the target language using the Google Translate API. We measure quality by back-translating the obtained examples back to English and measuring: (i) The *overlap* between the back-translated and the original English sentence, and (ii) The *delta* change in performance when PaLM2-S is evaluated on this back-translated version of GSM-8K as compared to the original version.

## A.2 GSM-8K

**Quality evaluation for LRL GSM-8K**  As described in Section 4.2, we created the GSM-8K dataset (Cobbe et al., 2021) for low-resource languages by using the Google Translate API to obtain silver translations in the target language from the source English sentence in the original dataset. We perform a quality evaluation of these examples by back-translating them back to English using the same translation API and defining two metrics over it:
(i) Overlap: The BLUE score measure between the actual example and the back-translated example,
(ii) Delta: The change in performance of the PaLM2-S model when evaluated on the original GSM-8K set as compared to the back-translated version.

Table 7 shows the values for these metrics across the various languages. We see that a decently high overlap value is seen across all languages. At the same time, the delta in performance is also minimal indicating that key attributes in the GSM-8K examples are not affected by translation.

**Results on the complete language set**  Table 6 shows the comparative evaluations on the complete set of 25 low-resource languages for which GSM evaluations are performed. We see an improvement over the anchor model $m_B$ for **20 of 25** languages. We also compare against the fully continued pre-trained version $m_B^{NTL}$ and observe that $m_{A \oplus B}$ outperform it for **18 of 25** languages.

## B  QUALITATIVE ANALYSIS

Table 8 depicts a few qualitative examples for the code-to-text, or the code explanation task, for Python. These examples depict examples for the three broader bucket of examples that we observe in cases when CALM yields the correct responses:

1. When neither of $m_A$ or $m_B$ generates the correct response but $m_{A \oplus B}$ correctly attends over their latent representations to yield the correct output,

2. When either of $m_A$ or $m_B$ is seen to give the correct response while the other one is incorrect and $m_{A \oplus B}$ generates the correct response that matches the generation from the correct model of $m_A$ and $m_B$, and

3. When both $m_A$ and $m_B$ generate the correct response and $m_{A \oplus B}$ reproduces those generations.

We also observed similar qualitative patterns with other tasks for language inclusivity.

## C  OVERHEAD WITH CALM

In this section, we include a detailed computation of the expected parametric and training overhead while composing given models using our proposed CALM framework.

<table>
<tr>
<td>

```python
def ConsumeBool(self):
  try :
    result = ParseBool(self.token)
  except ValueError as e :
    raise self._ParseError(str(e))
  self.NextToken()
  return result
```

</td>
<td>

```python
def value(self):
  if self.has_value:
    return self._impl[OBJ].get_val(K)
  else:
    raise ValueError("Not found")
  return
```

</td>
</tr>
</table>

| ⇒ Consumes a boolean | ⇒ Print an error message and exit. |
|---|---|
| $m_A$:    Consumes a boolean | [a part of the given model prefix] |
| $m_B$:    The object is not a member | Exit with error message |
| CALM:    Consumes a boolean | Print an error message and exit |

<table>
<tr>
<td>

```python
def get_positions(url):
  data = _get_resource(url)
  positions = [x for x in data['p']]
  return positions
```

</td>
<td>

```python
def distance(x0, y0, x1, y1):
  return (
    sqrt(pow(x1-x0,2) + pow(y1-y0,2)
  )
```

</td>
</tr>
</table>

| ⇒ Returns a list of positions. | ⇒ Returns the distance between two points |
|---|---|
| Positions of specified instruments. | Calculates the distance between two points |
| Get all positions. | Return the distance between two points |
| Returns a list of positions . | Calculates the distance between two points |

Table 8: Cherry-picked qualitative examples for the code-to-text task on Python that depict examples that fall into a set of larger bucket of patterns that we observe across examples. CALM does well in various settings: (i) when $m_A$ produces the correct output but not $m_B$, (ii) vice-versa—when $m_B$ does well, and (iii) when neither of the two base models do well but a combination of intermediate representations allow the composed model to give the correct output. This shows that composition implicitly learns to do both: routing across models and a combination, based on a given input.

## C.1 PARAMETRIC OVERHEAD

Building from the notations in §3.1, let's say the two models $\mathbf{m_A}$ and $\mathbf{m_B}$ have $N_A$ and $N_B$ number of standard transformer layers, respectively, with each layer of output dimensionality $D_A$ and $D_B$. As mentioned, we choose $n = n_A = n_B$ number of layers to perform the composition.

| | |
|---|---|
| # params per $f_{\text{proj}}$ layer | $= (D_A * D_B)$ |
| # params per $f_{\text{cross}}$ layer | $= (3 * D_B^2)$ |
| # total added params | $= n * (D_A * D_B + 3 * D_B^2)$ |
| # params in $\mathbf{m_B}$ | $= N_B * (V_B * D_B$ |
| | $\quad + 3 * D_B^2$ |
| | $\quad + 2 * D_B * D_B * K_B)$ |

where, $V_B$ and $K_B$ depict the vocabulary size and hidden multiplication factor, respectively.

Let's consider some standard transformer configurations to understand the parameter overhead. As an example, consider the layer configurations of standard BERT models: BERT-small ($\mathbf{m_A}$) and BERT-large ($\mathbf{m_B}$). That is: $N_A = 4$, $D_A = 512$, $N_B = 24$, $D_B = 1024$, $V_B = 30\text{K}$, $K_B = 4$. Assuming that we select all layers of $\mathbf{m_B}$, the value of $n = 4$. Then:

| | |
|---|---|
| # params for CALM | $= 4 * (512 * 1024$ |
| | $\quad + 3 * 1024^2)$ |
| | $\approx 1.5 \times 10^7 \approx 15\text{M}$ |
| # params in $\mathbf{m_B}$ | $= 24 * (30\text{K} * 1024$ |
| | $\quad + 3 * 1024^2$ |
| | $\quad + 2 * 1024^2 * 4) \approx 1\text{B}$ |
| %age added parameters | $= (15\text{M}/1\text{B}) * 100 = 1.5\%$ |

Hence, number of parameters added during composition is approximately 1.5% of those in $\mathbf{m_B}$.

## C.2 TRAINING OVERHEAD

Although learning the new parameters during CALM training requires back propagating over $\mathbf{m_B}$, the total training costs are still significantly lesser than fine-tuning $\mathbf{m_B}$, owing to the significantly lesser training examples.

Firstly, as discussed in the previous section, the additional number of parameters introduced during composition is ~1.5% of those in $\mathbf{m_B}$—hence, a negligible parametric addition. Further, since only ~5-7% of the total $\mathbf{m_B}$ fine-tuning data is required to train CALM, the training cost of CALM is minimal with respect to that of fine-tuning $\mathbf{m_B}$.

Let's assume that (i) the number of parameters in $\mathbf{m_B}$ is $P_B$ and the number of examples required to fine-tune $\mathbf{m_B}$ is $D_B$, (ii) the cost (in FLOPS), $C(.)$, of training scales linearly with parameters and data, i.e., $C(\mathbf{m_B}) = O(P_B * D_B)$ (iii) the number of parameters in $\mathbf{m_A}$ is 10% of those in $\mathbf{m_B}$, (iv) the number of parameters added for CALM is 2% of those on $\mathbf{m_B}$, and (v) the amount of data required to train CALM is 5% of $\mathbf{m_B}$ training. Then:

| | |
|---|---|
| $P_{\mathbf{m_{A \oplus B}}}$ | $= P_A + P_B + P_{\Theta_\mathbf{C}}$ |
| | $= 0.10 * P_B + P_B + 0.02 * P_B$ |
| | $= 1.12 * P_B$ |
| $D_{\mathbf{m_{A \oplus B}}}$ | $= 0.05 * D_B$ |

| | |
|---|---|
| $C(\mathbf{m_{A \oplus B}})$ | $= O(P_{\mathbf{m_{A \oplus B}}} * D_{\mathbf{m_{A \oplus B}}})$ |
| | $= O((1.12 * P_B) * (0.05 * D_B))$ |
| | $= O(0.056 * P_B * D_B)$ |
| | $= 5.6\% * O(P_B * D_B)$ |
| | $= 5.6\% * C(\mathbf{m_B})$ |

Hence, the cost of training CALM is a significantly lesser ($<10\%$) than that of fine-tuning $\mathbf{m_B}$.

