# OpenReview forum: "LLM Augmented LLMs: Expanding Capabilities through Composition"
_ICLR.cc/2024/Conference — ICLR 2024 poster_

### Official Review · Reviewer_AR8D · 2023-10-27

**Soundness:** 3 good
**Presentation:** 3 good
**Contribution:** 3 good
**Rating:** 6
**Confidence:** 4

**Summary:**

This paper proposes CALM to compose a general LLM with specialized models with a cross-attention module. By only fine-tuning the cross-attention on a small amount of data, the composed system is able to enable new capabilities for compositional tasks that neither of the two models can handle independently. The authors conduct experiments on tasks like math reasoning on low-resource language and code understanding and generation to verify the effectiveness of the proposed method.

**Strengths:**

- Though similar ideas are tested to be effective for building multi-modal models (e..g. reuse image encoder outputs for an LLM to enable multi-modal capabilities), this paper extends the application to new domains and scenarios.
- The method is simple and effective.

**Weaknesses:**

- Lack of important details, e.g. it is unclear how many the size of these models, the training steps, and hyperparameters. And how large is the training data $D_c$?
- It would be helpful to add a discussion on the composition with existing methods, such as routing and tool-using. If we treat the specialized models as external tools and prompt the general LLM to first call these models and then process the returned outputs, is it a kind of composition in the text space rather than representations?

**Questions:**

Please see above.

---

> ### Author Response · Authors · 2023-11-20
> **Response to Reviewer AR8D (1/2)**
>
> We thank the reviewer for their feedback. We are glad to see that the reviewer understands the usability of our work with respect to past approaches and appreciates the simplicity of our approach. We address their questions and comments below:
>
>
> > It would be helpful to add a discussion on the composition with existing methods, such as routing and tool-using
>
> Using models as tools and performing composition in the text space is definitely a very interesting direction. We have now added a discussion on this line of work in **Section 2**.
>
> Upon the reviewer's suggestion, **we conduct a experiment to compare with a routing approach in our key-value setup**. Particularly, we consider a much harder version of the task where the final **key-value arithmetic question is presented in the form of natural language text**: "What is the value when <key_1> is added to the difference of <key_2> and <key_3>"? In order to solve this task, a system would first need to parse the question in a simpler form, "<key_1> + <key_2> - <key_3>", then perform substitution of keys to values, and finally perform the arithmetic. We evaluate CALM on this hard task and compare with a skyline mA-to-mB routing performance by taking a product of mA's substitution performance on this task with the arithmetic performance of mB. We report the numbers here:
>
> |                            |      mA     | mB     |  CALM  |  Routing (best case) |
> |----------------------------|:-------------:|-------------:|-------:|-------:|
> | KV-Substitution                 |     98.1      |     0.0      |   92.9  | -- |
> | Numeric Arithmetic             |     4.2                |     73.7      |   72.0 | -- |
> | KV-Arithmetic (Natural Language Hard)            |     0.0  |     0.0            |     **36.7**       |     23.6  |
>
> We observe that **CALM outperforms the skyline routing performance by around 13 absolute accuracy points**.\
> This experiment demonstrates that CALM is much more effective for scenarios when **back-and-forth computation** needs be performed. In this case, the anchor model first needs to process the given input, which then requires the augmenting model's key-value mapping, and finally back to the anchor model for the arithmetic computation. Evidently, composition through representations allows this back-and-forth computation as well as the flexibility of employing either model (or some combination, thereof) at each decoding step.
>
> Further, in comparison to a framework such as **Toolformer** (Schick et al. (2023)) where a model is taught to call external tools during training, we believe **CALM presents several advantages**. The former requires the creation of a separate dataset to teach the model to use external tools (here, models), which in turn requires a large amount of prompt engineering and manual effort, before additional training. This is in contrast to CALM where we reuse the existing dataset for training and do not require any manual intervention. Moreover, composing via representations might also have the advantage of utilizing internal model computations, and hence might be more effective.
>
> Performing more targeted comparisons, towards this end, might be useful to compare the efficacy and training efficiency of the two approaches. While, setting up this setup and performing comparison with models as tools is beyond the scope of this rebuttal, we hope to include them in future versions of this work. Finally, we believe that both composition in the text space and that through internal representations have their individual strengths and might evolve to be useful across different use cases.

---

> > ### Author Response · Authors · 2023-11-20
> > **Response to Reviewer AR8D (2/2)**
> >
> > > It is unclear how many the size of these models, the training steps, and hyperparameters. And how large is the training data DC?
> >
> > Due to the proprietary nature of our models, we are unable to provide specific details regarding the models. However, for the experimental setting considered in our paper, the additional parameters introduced for composition are **less than 2% of the number of parameters in the anchor model**.
> >
> > Upon the reviewers’ suggestion, we include a detailed computation of the expected overhead assuming generic transformer models in **Appendix C.1**. We include a brief of the same here for the reviewer’s reference:
> >
> > Extending from the notations in Section 3.1, let's say the two models mA and mB have NA and NB number of standard transformer layers, respectively, with each layer of an output dimensionality of DA and DB. As mentioned, we choose n = nA = nB number of layers to perform the composition.
> > ```
> > # Parameters for each f_proj layer = (DA * DB)
> > # Parameters for each f_cross layer = (3 * DB**2)
> > # Total parameters introduced during composition = n * (DA * DB + 3 * DB**2)
> > # Parameters in mB = NB * (VB*DB + 3 * DB**2 + 2*DB*DB*KB)
> > ```
> > where, VB and KB depict the vocabulary size and hidden dimension multiplication factor, respectively.
> >
> > %age of new parameters added = (\# Total parameters introduced during composition\*100) / (\# Parameters in mB)
> >
> > Let's consider some standard transformer configurations to understand the parameter overhead.\
> > As an example, we consider the layer configurations of the BERT models: BERT-small (mA) and BERT-large (mB).\
> > In this case: NA = 4, DA = 512, NB = 24, DB = 1024, VB = 30K, KB = 4. Assuming that we select all layers of mA, n = 4.
> >
> > Hence,
> > ```
> > # Total parameters introduced during composition = 4  * (512 * 1024 + 3 *1024**2) ≈  1.5x107 ≈ 15M
> > # Parameters in mB = 24 * (30K*1024 + 3 * 1024**2 + 2*1024*1024*4) ≈ 1B
> > %age of new parameters added = 15M * 100 / 1B = 1.5%
> > ```
> > Hence,\
> > **Total parameters introduced during composition ~1.5% of parameters in mB**
> >
> > Further, for all our experiments, **DC is ~5-7% of the data used to train mA (or fine-tune mB)**.
> >
> >
> > ***
> >
> > We hope that the rebuttal clarifies the questions raised by the reviewer. We would be happy to discuss any further questions about the work, and would appreciate an appropriate increase in the score if the reviewer’s concerns are adequately addressed.

---

> > > ### Author Response · Authors · 2023-11-22
> > > **Follow-up**
> > >
> > > We would be delighted to discuss any further questions or clarifications regarding our work that haven't been addressed in the rebuttal.

---

### Official Review · Reviewer_6aS2 · 2023-11-02

**Soundness:** 4 excellent
**Presentation:** 3 good
**Contribution:** 4 excellent
**Rating:** 8
**Confidence:** 4

**Summary:**

The paper proposes CALM, composition to augment language models. CALM is designed to compose LLMs with different capabilities. In particular, the paper focuses on composing an anchor model with a domain-specific augmenting model to enable new capabilities. CALM introduces simple linear transformations to extract features from the augmenting model and insert them into the anchor model with cross-attention modules. In the experiments, the paper introduces three different applications of CALM, in the domains of key-value arithmetic, low-resource language inclusivity, and code understanding and generation. The experimental results demonstrate the composed LLMs can perform new challenging tasks and obtain better results than the models without composition.

**Strengths:**

- The idea of LLM composition is interesting. A great number of LLMs are designed with diverse capabilities. Combining LLMs with different capabilities and deriving new capabilities are valuable research questions. In section 4.2, after combining the model with KV-substitution skill and the model with numeric-arithmetic skill, the combined model achieves zero-shot KV-arithmetic inference. In comparison, the underlying models fail to do KV-Arithmetic inference, demonstrating the emergence of "new skills".
- The authors conduct extensive experiments on diverse domains ranging from language inclusivity to code generation. The results show that CALM successfully combines the capabilities of the anchor and augmenting models.

**Weaknesses:**

- Except for the key-value arithmetic case, the composition between low-resource languages and English, and the composition of coding and language capabilities are similar to the works of efficient cross-modal LLMs. For example, many studies have explored how to efficiently connect image encoders to LLMs, which finally achieves a "combined skill" like image-grounded language generation. I believe the relation between CALM and these related cross-modal models should be discussed. Besides, I would expect the combined models to have more non-trivial combined skills ( key-value arithmetic) beyond conditional generation (machine translation or code-to-text generation).
- Introduction mentions that directly training LLMs is computationally expensive but it seems that the experiments do not involve whether CALM is more computationally efficient than directly training the anchor model.

**Questions:**

Many studies have explored how to efficiently connect image encoders to LLMs, which finally achieves a "combined skill" like image-to-text generation. What is the difference between CALM and the efficient cross-modal LLM methods?

---

> ### Author Response · Authors · 2023-11-20
> **Response to Reviewer 6aS2**
>
> >  Many studies have explored how to efficiently connect image encoders to LLMs, which finally achieves a "combined skill" like image-to-text generation. What is the difference between CALM and the efficient cross-modal LLM methods? [...] I believe the relation between CALM and these related cross-modal models should be discussed.
>
> We have now expanded our discussion with respect to this body of past work in Section 2.
>
>
> > I would expect the combined models to have more non-trivial combined skills ( key-value arithmetic) beyond conditional generation (machine translation or code-to-text generation)
>
> In Table 3 we show CALM outperforms other models on MGSM tasks on NTL which we believe is a good evidence of combined skill (math reasoning in NTL languages). Similarly for code, we showcase superior performance for code completion in Table 4. We would be delighted to consider other potential scenarios and tasks to demonstrate non-trivial combined skills if the reviewer has some particular suggestions in this regard.
>
>
> > the experiments do not involve whether CALM is more computationally efficient than directly training the anchor model.
>
> The total training cost of training CALM is significantly lesser than training the anchor model.
> For all experiments reported in our work, CALM was trained only for 5-7% of the data as full mB training, and hence **training CALM is up to 10 times cheaper than training the anchor model**.
>
> Upon the reviewer’s suggestion, we now include a detailed discussion on training overhead with CALM in **Appendix C**.\
> For the reviewer’s reference, we find that:
>
> If we assume the number of parameters in mA to be 10% of those in mB and the amount of data required to train CALM is 5% of mB training.
>
> Assuming a linear scaling factor of training cost with model parameters and data:\
> **Cost of training CALM ~5% of the cost for training mB**\
> **Net cost of training mA + CALM ~15% of the cost for training mB**
>
> ***
>
> We would be happy to discuss any further questions about the work and would appreciate support for acceptance of the paper.

---

> > ### Comment · Reviewer_6aS2 · 2023-11-22
> >
> > Thank you for addressing my comments.

---

### Official Review · Reviewer_QMa2 · 2023-11-07

**Soundness:** 3 good
**Presentation:** 3 good
**Contribution:** 3 good
**Rating:** 6
**Confidence:** 3

**Summary:**

The paper introduces the CALM framework, which aims to enhance an existing LLM by incorporating a smaller, specialized one. This framework combines two pretrained LLMs while keeping their weights intact and utilizes learnable linear projections and self-attention to map intermediate representations from one model to the other. CALM demonstrates superior performance compared to both the anchor and augmenting models across three diverse tasks, including arithmetic reasoning, low-resource language translation, and code generation, which are not part of direct training but require a combination of the capabilities of both models.

**Strengths:**

1. Comprehensive Experiments: The authors conduct extensive experiments that span three widely used application domains. The choices of training and evaluation tasks are reasonable and effectively show the composition of two LLMs.

2. Performance Improvement: The experimental results consistently demonstrate performance improvements across all domains. Notably, the arithmetic reasoning experiment on synthetic datasets shows an impressive performance gain in comparison to both the anchor and augmenting models. The ability of CALM to tackle a novel task not previously feasible for either model is particularly intriguing.

3. Usefulness: The paper presents a concept with potential real-world applications. It enhances model reusability and eliminates the need to scale up existing models for injecting new knowledge.

4. Clarity: The paper is well-written and easy to follow.

**Weaknesses:**

1. Lack of Comparison to Existing Methods: While the idea of combining models with different skill sets has been explored in related work, such as ensembling hidden representations of two models, the paper lacks empirical results for such comparisons. Authors are recommended to consider providing a comparison of CALM with existing methods for ensembling model representations. Additionally, could the authors clarify "LoRA assumes access to the full underlying domain data during pretraining"? It is assumed that the same training data are used for both LoRA and CALM in Table 5.

2. Missing Information on Overhead: The paper does not provide information on the additional parameters introduced and their ratio to the anchor and augmenting models, which is important to understand the system's overhead. It would also be helpful to support the claim that "The composed model achieves similar performance for a tiny fraction of the training cost" with evidence, such as the number of FLOPs and memory requirements. Since the framework requires backpropagation through the initial layers of both models, I assume that the training cost should be comparable to directly training $m_B$.

3. The paper should include model configurations and training details to enhance reproducibility.

4. Some notations are used before being formally introduced, leading to potential confusion. For instance, notation $\mathbf{C}$ is introduced in Section 3.2 after being used from the beginning of Section 3. Additionally, both $t_{\{A, B\}}$ and $C$ denote task sets, which can be confusing and should be clarified or distinguished.

**Questions:**

1.  Given that CALM requires the model weights access and running both forward and backward passes, keeping the weights frozen isn't necessarily required. It would be interesting to investigate whether updating an anchor, augmenting, or both models could enhance performance.
2. It would be also interesting to explore the interactions between $m_B$ attending to $m_A$ in comparison to the current direction, to determine whether the capabilities of "generic" models can be effectively transferred to a "specialized" model, especially given that $m_A$ is more efficient to run.
3. Table 3 shows more noticeable performance gains in high-resource languages compared to low-resource ones. I wonder if this might imply that the composition strengthens the skills of pretrained models thanks to extra parameters, rather than teaching them entirely new skills.

---

> ### Author Response · Authors · 2023-11-18
> **Response to Reviewer QMa2 (1/3)**
>
> We thank the reviewer for their careful analysis of our paper and the extremely insightful comments. We are glad to that they find our work useful and appreciate the comprehensiveness of our experiments. We really appreciate the clarity of the questions raised by them. We address the reviewer's comments below:
>
>
> > The paper does not provide information on the additional parameters introduced and their ratio to the anchor and augmenting models, which is important to understand the system's overhead
>
> Due to the proprietary nature of our models, we are unable to provide specific details regarding the models. However, for the experimental setting considered in our paper, the additional parameters introduced for composition are **less than 2% of the number of parameters in the anchor model**.
>
> Upon the reviewers’ suggestion, we include a detailed computation of the expected overhead assuming generic transformer models in **Appendix C.1**. We include a brief of the same here for the reviewer’s reference:
>
> Extending from the notations in Section 3.1, let's say the two models mA and mB have NA and NB number of standard transformer layers, respectively, with each layer of an output dimensionality of DA and DB. As mentioned, we choose n = nA = nB number of layers to perform the composition.
> ```
> # Parameters for each f_proj layer = (DA * DB)
> # Parameters for each f_cross layer = (3 * DB**2)
> # Total parameters introduced during composition = n * (DA * DB + 3 * DB**2)
> # Parameters in mB = NB * (VB*DB + 3 * DB**2 + 2*DB*DB*KB)
> ```
> where, VB and KB depict the vocabulary size and hidden dimension multiplication factor, respectively.
>
> %age of new parameters added = (\# Total parameters introduced during composition\*100) / (\# Parameters in mB)
>
> Let's consider some standard transformer configurations to understand the parameter overhead.\
> As an example, we consider the layer configurations of the BERT models: BERT-small (mA) and BERT-large (mB).\
> In this case: NA = 4, DA = 512, NB = 24, DB = 1024, VB = 30K, KB = 4. Assuming that we select all layers of mA, n = 4.
>
> Hence,
> ```
> # Total parameters introduced during composition = 4  * (512 * 1024 + 3 *1024**2) ≈  1.5x107 ≈ 15M
> # Parameters in mB = 24 * (30K*1024 + 3 * 1024**2 + 2*1024*1024*4) ≈ 1B
> %age of new parameters added = 15M * 100 / 1B = 1.5%
> ```
> Hence,\
> **Total parameters introduced during composition ~1.5% of parameters in mB**
>
>
> > Since the framework requires backpropagation through the initial layers of both models, I assume that the training cost should be comparable to directly training mB.
>
> The reviewer is correct that the training FLOPs are comparable for both mB and CALM for a fixed set of examples since the gradients are computed over mB during backpropagation. However, the training cost of CALM is significantly smaller than the anchor model because it is trained over a very small fraction of the total training examples and iterations as full anchor model training.
>
> For all experiments reported in our work, CALM was trained only for 5-7% of the data as full mB training, and hence **training CALM is up to 10 times cheaper than training the anchor model**.
>
> We now include a detailed discussion on training overhead with CALM in **Appendix C.2**. We include salient parts of the details here for the reviewer’s reference:
>
> Firstly, as discussed above, the additional number of parameters introduced during composition is <2% of the number of parameters of mB—hence, a negligible addition in training cost per example.\
> Further, since only ~5-7% of the total mB fine-tuning data is required to train CALM, the training cost of CALM is minimal with respect to training cost of training the entire anchor model.\
> Moreover, since our experiments consider an mA that has 5-20% of parameters as mB, even the net cost of training mA and CALM is significantly lesser than training mB.
>
> Let’s say that the number of parameters in mA is 10% of those in mB and the amount of data required to train CALM is 5% of mB training.
>
> Assuming a linear scaling factor of training cost with model parameters and data:\
> **Cost of training CALM ~5% of the cost for training mB**\
> **Net cost of training mA + CALM ~15% of the cost for training mB**

---

> ### Author Response · Authors · 2023-11-18
> **Response to Reviewer QMa2 (2/3)**
>
> > While the idea of combining models with different skill sets has been explored in related work, such as ensembling hidden representations of two models, the paper lacks empirical results for such comparisons
>
>
> The existing methods for ensembling model representations might not be directly comparable with CALM for the following reasons (a) The augmenting model(s) mA and anchor model mB can be of different size and their representation dimension can differ, which are not suitable for representation ensembling models. (b) The ensemble models work best when the original models are well aligned i.e. they are derived from the same base model, CALM is applicable for any set of models.
>
>
> On the reviewer’s suggestion, we have now expanded our evaluations with other baselines such as efficient fine-tuning approaches like LoRA for all our tasks and setups (NTL and Code). We report these comparisons in Table 5 and present a snippet here:
>
>
> |                            |      CALM     |  LoRA  |
> |----------------------------|:-------------:|-------:|
> | HumanEval [pass@1]                 |     22.5      |   18.3 |
> | MBPP [pass@1]                       |     32.2      |   28.7 |
> | FLORES-200 [#( >mB)]                    |     175       |     82 |
> | GSM 8K (LRL) [#( >mB)]                    |      20       |     15 |
>
>
> We observe CALM outperforms LoRA in most of the tasks across NTL and code tasks.
>
>
>
>
> > Table 3 shows more noticeable performance gains in high-resource languages compared to low-resource ones. I wonder if this might imply that the composition strengthens the skills of pretrained models thanks to extra parameters, rather than teaching them entirely new skills.
>
> That's an important question and the set of ablation experiments in the paper specifically address this question.
> In order to see whether the performance gains through composition occur due to the utilization of knowledge from mA or from the additional parameters, we replaced mA with randomly initialized versions.
>
> |                            | CALM | Vanilla mA | Random mA |
> |----------------------------|------|------------|-----------|
> | FLORES-200 [avg. acc.]   | 60.5 | 59.2       | 58.8      |
> | #(>mB)                     | 175  | 115        | 43        |
> | GSM 8K (LRL) [avg. acc.]          | 21.4 | 19.0       | 17.8      |
> | #(>mB)                    | 20   | 15         | 9         |
> | GSM 8K (HRL) [avg. acc.]          | 33.1 | 29.7       | 28.5      |
> | #(>mB)                     | 11   | 8          | 4         |
>
> Table 5 (a snippet shown above) shows the corresponding results. We observed that both "vanilla" (mA replaced with a vanilla pre-trained checkpoint) and "random" (a randomly set of mA weights) show significant decrement in performance. For all these cases, the number of additional parameters introduced remains the same. Hence, **the difference in performance is reflective of the utility of mA through composition**.
>
> Moreover, the rank for LoRA is chosen such that the same number of parameters are added. The inferior performance therein further supports our claims that composition with mA is useful.
>
>
> > Some notations are used before being formally introduced, leading to potential confusion.
>
> We thank the reviewer for their careful observation and for pointing this out. We have now updated the notations and have made the surrounding descriptions more clear.

---

> ### Author Response · Authors · 2023-11-18
> **Response to Reviewer QMa2 (3/3)**
>
> >  Could the authors clarify "LoRA assumes access to the full underlying domain data during pretraining"? It is assumed that the same training data are used for both LoRA and CALM in Table 5.
>
>
> Apologies for the confusion. Here we are alluding to a potential proprietary situation where the full set of downstream domain data is unavailable at the time of adaptation, but the entire domain data has been parametrically condensed in an augmenting model that is available to us. We have observed that training CALM over an available data subset suffices in augmenting the desired set of knowledge to an anchor model. However, this is not possible with LoRA since the parameters are only exposed to the data subset available during training.
>
> As an example, our key-value arithmetic setup involves a set of 25K keys. We performed CALM training only over a held-in set of 5K key-value pairs and evaluated over the remaining 20K held-out keys (Table 1).
>
> We have now made this more clear in the paper.
>
>
> > Given that CALM requires the model weights access and running both forward and backward passes, keeping the weights frozen isn't necessarily required. It would be interesting to investigate whether updating an anchor, augmenting, or both models could enhance performance
>
> We agree. It would be interesting to investigate whether updating the models during composition could enhance performance. It is possible that updating the augmenting models during this process leads to better alignment with the anchor model.
>
> However, in this work, we inherently assume a setup where the underlying models are given upfront and updating their parameters is not permitted. This setup is of practical importance in several scenarios such as when serving a proprietary large model to downstream customers/users who need to utilize the model for their custom knowledge.
>
> Further, keeping the given models unchanged ensures control for catastrophic forgetting (as evidenced by our results on GSM-8K and CodeXGLUE). Changing model parameters can lead to undesirable and intractable changes.
>
> A setting where we update only very few layers in the two models could be interesting to explore and we can investigate this going forward. Since this exploration requires large experimental bandwidth, we include it as a part of our future plans.
>
> > It would be also interesting to explore the interactions between mB attending to mA in comparison to the current direction, to determine whether the capabilities of "generic" models can be effectively transferred to a "specialized" model, especially given that mA is more efficient to run.
>
> That's an interesting suggestion. However, an implicit directionality is assumed in how we define augmenting and anchor models.
> By definition, the anchor model has coherent text generation attributes (along with other higher level capabilities such as reasoning) but misses some specialized skills (such as domain knowledge) that, in turn, is present in the augmenting models. Once the relevant skills are augmented through composition, decoding from the same anchor model follows the logical ordering in the framework.
>
> Reversing this ordering might be useful for certain settings (for instance where we wish to restore specialized model's performance on a task that it has forgotten) and investigating such scenarios could be very interesting, however beyond the scope of this rebuttal.
>
> ***
>
> We would be happy to discuss any further questions about the work, and would appreciate an appropriate increase in the score if the reviewer’s concerns are adequately addressed.

---

> > ### Author Response · Authors · 2023-11-22
> > **Follow-up**
> >
> > We would be delighted to discuss any further questions or clarifications regarding our work that haven't been addressed in the rebuttal.

---

### Official Review · Reviewer_kQBq · 2023-11-10

**Soundness:** 2 fair
**Presentation:** 2 fair
**Contribution:** 3 good
**Rating:** 6
**Confidence:** 4

**Summary:**

The paper proposes a new approach, CALM, to compose an anchor model and a domain-specific augmenting model by introducing a small number of trainable parameters over both models’ intermediate layer representations. This approach combines the capabilities of the anchor model and the augmenting model, and therefore, is able to address new challenging tasks that cannot be solved by either model alone. The authors use three tasks to demonstrate the advantages of CALM:
1. solving arithmetic expressions containing keys: the anchor model is trained to do arithmetic over integers, and the augmenting model is trained to memorize string-to-integer key-value mappings.
2. Translation and math-word problem-solving in low-resource languages: the anchor model is a pretrained PaLM-2 S model and the augmenting model is trained on low-resource languages.
3. Code completion, text-to-code generation, and code-to-text generation: the anchor model is a pretrained PaLM-2 S model and the augmenting model is pretrained on codes.

**Strengths:**

Strengths
1. The new approach enables new capabilities in the composed version that the original models cannot achieve.
2. CALM does not modify the parameters of the original model, avoiding the catastrophic forgetting that is prevalent in conventional approaches.
3. CALM has a flexible architecture that makes it possible to compose more than one augmenting model with an anchor model.

**Weaknesses:**

1. The first task, key-value arithmetic, seems rather arbitrary. A simple encoder-decoder model is expected to solve this problem, leaving the readers the question of why they bother to use the composition of two models.
2. For the second task, machine translation in low-resource languages, when the anchor model is trained on the low-resource languages, its performance is higher than the composed approach, which again raises the question of why you need to compose two models instead of fine-tuning the anchor model.
3. The paper misses the details of parameterization/hyperparameter selection. For example, the authors did not write how the layers of the anchor and the augmenting models are selected.

**Questions:**

1. It is not clear to me how the projection function works. Does it project each selected layer of A to all selected layers of B? What is the definition of HBj? Does HA⊕Bj contain the information from all HA or only from a specific layer HAi?
2. Why does CALM on the KV-Arithmetic task have a higher score than mB on the  Numeric-Arithmetic task? Should mB trained on the Numeric-Arithmetic task be the upper bound?

---

> ### Author Response · Authors · 2023-11-20
> **Response to Reviewer kQBq (1/2)**
>
> > The first task, key-value arithmetic, seems rather arbitrary. A simple encoder-decoder model is expected to solve this problem, leaving the readers the question of why they bother to use the composition of two models.
>
> While encoder-decoder models are capable of solving arithmetic tasks, our key-value arithmetic setup is **rather unique** where it aims to evaluate whether expert knowledge of one model can be transferred into general capabilities of an anchor model.
>
> In this setup, all key assignments are stored in an expert model mA (var1=4, var2=5). While mB has the capability to do arithmetic (4+5=9), it is incapable of performing arithmetic on keys (var1+var2=?). The CALM framework aims to transfer the key-assignment knowledge from mA into mB and demonstrates that the composed model is able to perform arithmetic on keys (var1+var2=9).
>
> Below we will explain the exact setup of our task-
> (i) All key assignments, {var1: 4, var2: 5, ..., varN: X}, are stored in mA,
> (ii) We perform CALM training with mA and mB on a certain number of **held-in keys**, {var1, var2, ..., varH}, where H << N. As mentioned in Section 4.1, we use 20% of the keys stored in mA during CALM training.
> (iii) We evaluate key-value arithmetic performance on **held-out keys not seen during CALM training**: {varH+1, ..., varN}.
>
> Happy to discuss more if we misunderstood your question or if you would like further clarification.
>
> > For the second task, machine translation in low-resource languages, when the anchor model is trained on the low-resource languages, its performance is higher than the composed approach, which again raises the question of why you need to compose two models instead of fine-tuning the anchor model
>
> Composition through CALM has significant advantages over fine-tuning:
>
> (i) CALM provides an easy and efficient way of adding specific knowledge to a general model and expanding capabilities of an anchor model. We have added a discussion in Appendix C discussing the cost of training CALM v/s that of fine-tuning the anchor model. We find that training through **CALM is up to 10 times cheaper than direct fine-tuning**. Fine-tuning of a large model, as considered in our experiments, is expensive. **For most of our experiments, we see comparable or better performance with composition for a fraction of the cost**.
>
> (ii) While fine-tuning performance for the low-resource translation task exceeds that of CALM, **Table 3 provides evidence of forgetting due to fine-tuning**. We see that fine-tuning led to regression on numeric arithmetic on both high and low-resource languages (performing equal or lower than the base anchor model) while CALM still performs well, exceeding both fine-tuned and base versions. We provide a summarized snippet of the results below for the reviewer's reference:
>
> |                            |      Base anchor     | CALM     |  Fine-tuned  |
> |----------------------------|:-------------:|-------------:|-------:|
> | FLORES-200                 |     41.7      |     46.1      |   **48.2**  |
> | GSM-8K (LRLs avg.)             |     22.5                |     **25.0**      |   22.6 |
> | GSM-8K (HRLs avg.)            |     28.0              |     **31.8**       |     25.6  |
>
> We also see similar findings for the code setup where the fine-tuned model on code (mostly involving code as output) shows large regression on the code-to-text tasks since they involve generating text output:
>
> |                            |      Base anchor     | CALM     |  Fine-tuned  |
> |----------------------------|:-------------:|-------------:|-------:|
> | HumanEval (code completion)                 |    16.4      |    22.5      |   **24.3**  |
> | CodeXGLUE (code-to-text; avg. over 6 languages)             |        32.33            |     **32.76**      |   28.98 |
>
>
> > The paper misses the details of parameterization/hyperparameter selection. For example, the authors did not write how the layers of the anchor and the augmenting models are selected.
>
> Our framework only involves one hyperparameter of n, specifying the number of layers used for composition in both models. For some value of n, the output representation from every (NA/n)th layer in the augmenting model is used for composition with every (NB/n)th layer in the anchor model  (where NA and NB are the number of layers in the two models, respectively).
>
> For all our experiments, we set the value of (NA/n) as 4. We have now specified this hyperparameter choice in Section 4.
>
>
> > How the projection function works. [...] What is the definition of DBj? Does DA⊕Bj contain the information from all DA or only from a specific layer DAi?
>
> The projection is a simple learnable linear layer that maps representations from mA to the dimensionality of mB.

---

> > ### Author Response · Authors · 2023-11-20
> > **Response to Reviewer kQBq (2/2)**
> >
> > > Why does CALM on the KV-Arithmetic task have a higher score than mB on the Numeric-Arithmetic task? Should mB trained on the Numeric-Arithmetic task be the upper bound?
> >
> > The anchor model (mB) was not trained on the numeric arithmetic task. Since it is a relatively large model, the base model itself has the arithmetic capability that we utilize during composition. When training CALM, the key-value arithmetic examples consist of chain-of-though reasoning involving numeric arithmetic, and hence a performance greater than mB's arithmetic performance is seen.
> >
> > As suggested by the second part of the reviewer's question, we will add a skyline for math numeric performance after training mB on the same data, which would be a true upper-bound for the key-value arithmetic performance.
> >
> >
> > ***
> > We hope that the rebuttal clarifies the questions raised by the reviewer. We would be happy to discuss any further questions about the work, and would appreciate an appropriate increase in the score if the reviewer’s concerns are adequately addressed.

---

> > > ### Author Response · Authors · 2023-11-22
> > > **Follow-up**
> > >
> > > We would be delighted to discuss any further questions or clarifications regarding our work that haven't been addressed in the rebuttal.

---

### Author Response · Authors · 2023-11-20
**General Response**

We thank the reviewers for their thoughtful and insightful reviews.  We are glad to learn that the reviewers find the idea of composing LLMs for augmenting new knowledge exciting and appreciate the simplicity, flexibility, and efficiency of our approach. We are happy to see that the reviewers appreciate the soundness of our work and find our experimental setups comprehensive and convincing.

Based on reviewers’ recommendation, we have added a thorough discussion on the parametric and training overhead with CALM in Appendix C. Overall, we note that CALM introduces as low as **2% of the anchor model parameters** (Appendix C.1). Further, CALM training is performed on ~5-7% of the total training set (on which the augmenting model training is performed). On collating these numbers (Appendix C.2), we find that training via CALM is **up to 10 times cheaper** than directly training the anchor model on domain data.

We have elaborated our discussions in the related works section (Section 2) to accommodate reviewers’ suggestions regarding some prior work. We have also performed more comprehensive comparisons with ablations and report strong performance differences in Table 5.

We address other feedback in response to the individual reviewer comments.

We list other changes made in the paper draft here:
- Code results
    - Better mA: Found an improved augmenting model by further pretraining PaLM2-XXS for more iterations and lower learning rate. This mA shows superior performance on different evals. Table 4 is updated accordingly.
    - Ablations: Reported comparative performance on replacing mA with vanilla PaLM XXS and random mA model. We also report LoRA performance where we use the same number of additional parameters and training data as CALM.
    - Added a full fine-tuning baseline (mBCode) in Table 4.
    - Performed model selection (across various sampling temperatures) and reported the best numbers for all models.
- Clarity, presentation, and notation.
    - Updated various notations for enhanced clarity in Section 3 upon reviewers’ suggestions.
    - Enhanced clarity of training data, model parameters. Made changes in the respective sections of the draft.
    - Included hyperparameter and reproducibility details in Section 4.

---

### Meta-Review · Area_Chair_WFEM · 2023-12-12

**Metareview:**

**Paper Summary:**

This paper introduces Composition to Augment Language Models (CALM), a method that augments large language models (LLMs) by composing them with domain-specific models. Designed to expand the capabilities of foundational LLMs without altering their original weights, CALM employs cross-attention mechanisms between models to facilitate new capabilities. The effectiveness of this approach is demonstrated through experiments in various domains.

**Strengths:**

1. Capability Enhancement: CALM enhances the capabilities of existing models without modifying their original parameters, thus preserving their initial strengths (kQBq, 6aS2).

**Weaknesses:**

1. Violation of Anonymous Policy: The first page of this paper includes the names and email addresses of two authors, violating the anonymous submission policy.
2. Novelty Compared to Existing Methods: The proposed approach is similar to and should be compared with existing methods that combine multiple models (QMa2, 6aS2), such as cross-modal LMs exemplified by "[Encoder-Agnostic Adaptation for Conditional Language Generation](https://arxiv.org/abs/1908.06938)".
3. Reproducibility Concerns: The paper lacks detailed information about model configurations and training, raising reproducibility concerns. The authors mention in their rebuttal, "Due to the proprietary nature of our models, we are unable to provide specific details regarding the models" (kQBq, QMa2).


**Decision:**

While the reviews suggest that this paper merits acceptance based on its content, it violates the anonymous policy by including author names and emails in the main paper. Consequently, I will temporarily recommend rejection for this paper and will consult with the SAC for further discussion.

**Justification For Why Not Higher Score:**

This paper violates the conference's anonymous policy.

**Justification For Why Not Lower Score:**

N/A

---

### Decision · Program_Chairs · 2024-01-16

Accept (poster)